# Post-Infectious Acute Cerebellar Ataxia Treatment, a Case Report and Review of Literature

**DOI:** 10.3390/children10040668

**Published:** 2023-03-31

**Authors:** Emanuela Del Giudice, Filippo Mondì, Greta Rachele Bazzanella, Alessia Marcellino, Vanessa Martucci, Giovanna Pontrelli, Mariateresa Sanseviero, Piero Pavone, Silvia Bloise, Salvatore Martellucci, Anna Carraro, Flavia Ventriglia, Miriam Lichtner, Riccardo Lubrano

**Affiliations:** 1Department of Maternal Infantile and Urological Sciences, Santa Maria Goretti Hospital, Sapienza University of Rome, Polo Pontino, 04100 Latina, Italy; 2Department of Maternal Infantile and Urological Sciences, Sapienza University of Rome, Viale del Policlinico 155, 00161 Rome, Italy; 3Department of Clinical and Experimental Medicine, University Hospital “Policlinico-San Marco”, 95123 Catania, Italy; 4Otorhinolaryngology Unit, Santa Maria Goretti Hospital, ASL Latina, 04100 Latina, Italy; 5Infectious Diseases Unit, Santa Maria Goretti Hospital, Sapienza University of Rome, Polo Pontino, 04100 Latina, Italy

**Keywords:** cerebellar incoordination, ataxia, infectious mononucleosis, intravenous immunoglobulin therapy

## Abstract

Background: infectious mononucleosis is very common during childhood and neurological manifestations are extremely rare. However, when they occur, an appropriate treatment must be undertaken to reduce morbidity and mortality as well as to ensure appropriate management. Methods: we describe the clinical and neurological records of a female patient with post-EBV acute cerebellar ataxia, whose symptoms rapidly resolved with intravenous immunoglobulin therapy. Afterwards, we compared our results with published data. Results: we reported the case of an adolescent female with a 5-day history of sudden asthenia, vomiting, dizziness, and dehydration, with a positive monospot test and hypertransaminasemia. In the following days, she developed acute ataxia, drowsiness, vertigo, and nystagmus with a positive EBV IgM titer, confirming acute infectious mononucleosis. The patient was clinically diagnosed with EBV-associated acute cerebellitis. A brain MRI showed no acute changes and a CT scan showed hepatosplenomegaly. She started therapy with acyclovir and dexamethasone. After a few days, because of her condition’s deterioration, she received intravenous immunoglobulin and demonstrated a good clinical response. Conclusions: although there are no consensus guidelines for the treatment of post-infectious acute cerebellar ataxia, early intervention with intravenous immunoglobulin might prevent adverse outcomes, especially in cases that do not respond to high-dose steroid therapy.

## 1. Introduction

Acute cerebellar ataxia (ACA) is a clinical syndrome characterized by the sudden onset of gait disturbance as a result of multiple causes, including life-threatening conditions such as neoplastic or vascular lesions, intoxications, central nervous system infections as well as self-limited etiologies, the most frequent of which occur post-infections [1,2].

However, ACA is usually a self-limited syndrome and numerous infectious agents have been implicated in its pathogenesis, including coxsackievirus, echovirus, enteroviruses, Epstein–Barr virus, hepatitis A, herpes simplex virus 1, human herpes virus 6, measles, mumps, parvovirus B19, Borrelia burgdorferi and Mycoplasma pneumoniae. It is a diagnosis of exclusion that usually develops a few days or weeks after a viral illness [1]. Recent infection with Epstein–Barr virus (EBV) associated with ataxia has been reported in older children and adolescents, although it is a rare complication [3,4,5]. The pathogenic mechanisms that underline acute cerebellar ataxia have not been completely defined, but autoimmune mechanisms seem to be involved [6,7]. Ataxia is considered a benign and self-limiting condition, but symptoms could persist for several weeks, with a negative impact on the patients’ quality of life. Prospective clinical trials on treatments are lacking and there are only few reports describing treatment with intravenous immunoglobulin (IVIG) in children [3,8,9].

We report the case of a 16-year-old patient with post-EBV infection ACA, whose symptoms rapidly resolved with IVIG therapy.

## 2. Case Presentation

A sixteen-year-old female was admitted to our Emergency Department for the acute onset of asthenia, vomiting, loss of appetite, dizziness, and dehydration. In the emergency room, she performed laboratory exams and ultrasound examination, showing leukocytosis (8870 × 10^3^ /µL leukocytes with lymphocytes 70.2%), hypertransaminasemia (aspartate transaminase 151 U/L, alanine aminotransferase 327 U/L) and elevated serum levels of gamma glutamyl transferase (124 U/L), lactate dehydrogenase (619 U/L), and C-reactive protein (0.54 mg/dL). Infectious screening was performed, including serologic tests (IgM and IgG) for CMV and toxoplasmosis that resulted negative; as well as a virologic test to confirm an EBV infection, and the results showed negative for the EBV nuclear antigen, but positive for the IgG and IgM viral capsid antigens; and a monospot test that resulted positive. Since the patient was examined during the pandemic and neurological symptoms are also present in SARS-infected children, antigenic and molecular swabs for SARS-CoV-2 were performed, as in all patients admitted to the ward following our Hospital protocol, which resulted negative [10]. The abdomen ultrasound showed liver enlargement with preserved morphology, without ascites, splenomegaly, gallbladder hyperechogenicity, and one reactive lymph node in the hepatic hilum.

Therefore, the patient was admitted to our Pediatric Department. She had no remote pathological history of diseases or notable family disorders. She was immunocompetent and regularly vaccinated according to the Italian health authorities, and had received an HPV vaccine two weeks before the onset of ACA. She had not yet been vaccinated against SARS-CoV-2 in accordance with the recommendations of Italian Ministry of Health, which were authorized shortly afterwards in June 2021 [11].

Moreover, no history or recent use of drugs was reported. At admission, she did not show focal neurological deficits, although a positive Romberg test had been highlighted, expressed by the tendency to fall with closed eyes in a holding position; furthermore, she had cervical and inguinal lymphadenopathy, mild hyperemic pharynx, palpable liver and spleen. The rest of the physical examination was unremarkable and vital signs were all normal for her age.

Thus, according to laboratory exams and the patient’s history of acute symptoms with nausea, dizziness and a positive Romberg test, a lumbar puncture and instrumental examinations were arranged. The examination of cerebrospinal fluid revealed 4 white cells per mm^3^ (reference range 0–5 cells/ mm^3^), 80 mmol/L glucose (reference range 60–70 % blood glucose concentration), and 18.9 mg/dL protein (reference range 15–45 mg/dL). The multiplex PCR Film Array for Meningitis/Encephalitis panel in cerebrospinal fluid was negative for bacteria and enterovirus, herpes simplex including EBV, HSV1-2, HHV-6 and mycoplasma pneumoniae. Additionally, anti-N-methyl-D-aspartate receptor antibodies and oligoclonal bands were not detected in her cerebrospinal fluid and blood. Moreover, magnetic resonance imaging (MRI) of the spine and brain showed no evidence of signal intensity or mass effect.

Two days after hospital admission, the patient’s condition started to deteriorate, characterized by drowsiness, persistent vomiting, dizziness associated with ataxia, unsteadiness of gait, and a positive Romberg test. To investigate the worsening of the neurological symptoms, we performed an electroencephalogram (EEG) that showed unspecific signs of theta dysrhythmia and a medium-posterior polymorphic aspect. Afterwards, an ophthalmological counseling with fundus examination resulted normal, while a spontaneous, primary position, horizontal jerk nystagmus with alternating fast phase direction (Periodic Alternating Nystagmus-PAN) was recorded during the videonystagmoscopy (an additional movie file shows this in more detail [see the Additional file in the Appendix A]). Therefore, a diagnosis of encephalitis was presumed, and the patient was treated with acyclovir at a dose of 15 mg/kg × 3 times/day for two days until the EBV serology test response was available, and dexamethasone at a dose of 0.25 mg/kg/day.

During the following days, the patient complained of the same symptoms with no improvement (persistence of drowsiness, ataxia, loss of strength in legs with trouble maintaining a standing position); therefore, we decided to start intravenous immunoglobulin therapy at a dose of 0.4 g/kg for 5 days and increase the dexamethasone dose to 0.4 mg/kg/day, according to the patient and her family. After 4 days of immunoglobulin therapy, the patient’s clinical condition improved in terms of dizziness and nausea, and she had normal neurological reflexes and was able also to stand up and start physical therapy. Therefore, intravenous immunoglobulin therapy showed a significant impact on the patient’s quality of life, improving her ability to resume daily activities and relieving distress in her parents.

Abdomen, thoracic and skeletal computerized tomography (CT) performed on the 7th day of hospitalization showed liver enlargement and splenomegaly, with preserved morphology, and some subcentimetric lymph nodes. A brain and spinal MRI on the 12th day of hospital admission was negative and laboratory tests were normal; therefore, she stopped acyclovir therapy. Because of the patient’s improved clinical condition, intravenous dexamethasone was progressively reduced and replaced with oral prednisolone.

On the 24th day of hospitalization, the patient was discharged with a normal neurological examination and no more signs of the disease. Prednisolone and home physiotherapy were prescribed, and she had a follow-up in our multidisciplinary outpatient clinic for about 1 year with full recovery.

## 3. Discussion

ACA is a clinical syndrome that is usually benign and transient, although life-threatening conditions could occur and may require immediate interventions.

ACA may affect people of all ages, but is usually seen in children under six years of age [12], although cases in older children and adolescents have been reported [3,8], often presenting as a post-infectious disorder [12]. The classical manifestations are gait disturbance and nystagmus, while associated symptoms may include dysarthria, vomiting, irritability or headache, seizures, and even alterations of consciousness can be present [12,13,14].

The most common causes of acute-onset ataxia are known to be drug ingestions, vaccination, viral or bacterial infections, malignancies and intoxications to lead, mercury, alcohol or ethylene glycol [14]. Several causal events may be pointed out to explain the onset of the acute cerebellar ataxia in our patient, including a possible trigger due to HPV vaccination carried out two weeks before the onset of clinical symptoms.

However, the clinical presentation, the results of laboratory exams, positive IgG and IgM viral capsid antigen serologic tests, as well as a positive monospot test, we inferred a temporal and suggestive causal association between EBV infection and acute cerebellar ataxia. Even though acute cerebellar ataxia is a rare complication of infectious mononucleosis [5], it could represent the only clinical manifestation [15].

Although HPV vaccination cannot be excluded as a possible cause of the temporal relationship with ataxia according to causality assessment criteria known [16,17], the findings are more likely explained by its facilitation of an amplified immunological response to the confirmed EBV infection [18].

A wide range of neurological manifestations has been described during EBV infection in pediatric and adult patients, whose main features are reported in Table 1. They include mild symptoms such as headache and drowsiness or more severe manifestations including seizures, ataxia or encephalitis [3,19,20,21,22,23,24,25,26,27,28,29,30,31,32,33,34,35,36].

In our case, we did not detect EEG and MRI abnormalities, but we found acquired periodic alternating nystagmus, which is considered by many authors to be a sign of cerebellar nodule or uvula dysfunction [37,38]. Some studies have reported specific brain damages such as cerebral edema, cortical lesions, and cerebral hemorrhages [21,22,23,24,25,26,28,29,30,31,32,33,36]. However, only one such patient has died [34]. Moreover, even though some cases of positive EBV CRP tests in cerebrospinal fluid have been described [15,39], the film array performed using our patient’s cerebrospinal fluid was negative for EBV.

Several studies suggest an autoimmune process [7,40,41], where ataxia is usually a self-benign manifestation driven by an immunomediated mechanism triggered by the viral infection [14,42]. These observations suggest that therapies directed at the immune system might be useful in treating post-EBV cerebellar ataxia. In fact, the clinical improvement in our patient occurred only after the administration of IVIG. As in other patients that did not respond to high-dose steroid therapy, IVIG therapy has been used in acute cerebellar ataxia, with good outcomes [3,8,9]. Therefore, the resumption of the patient’s daily activities and the alleviation of distress in her parents underline the importance of focusing on the quality of life aspect in order to improve the quality of medical care, as has also been highlighted in other diseases [43,44].

Immunological treatments are the most promising tools to achieve clinical remission and improve quality of life in several diseases [45]. These choices also show promise from a healthcare system spending perspective [46].

The main limitation of our work is the description of a single case report. Large controlled clinical trials administering only intravenous immunoglobulin are needed before definitive conclusions can be reached.

## 4. Conclusions

Acute cerebellar ataxia is considered a self-limiting condition and the prognosis is generally good, and it is usually treated conservatively. However, because symptoms can persist for months and there are also reports of patients with permanent residual motor deficits, as well as behavioral and cognitive issues, an early treatment directed at the immune response might decrease the duration and severity of symptoms, improving the outcome.

Prospective and multicenter studies involving a multispecialty team are necessary to confirm the best therapeutic approach.

## Figures and Tables

**Table 1 children-10-00668-t001:** Main characteristics related to neurological involvement during EBV infection in published pediatric and adult patients.

Study	Age/Sex	General	Neurologic	EEG	MRI	Leu/mL	Prot mg/dL	Glu mg/dL	EBV PCR
Cho TA [3]	19/f	Fe, Fa, L, O, Ph	A, Dy, Ga, Ny,	NC	Neg	4	28	53	Neg
Hussain RS [19]	25/f	Fe, Fa, L Na, Ph	A, D, Dy, Ga, He, N	NC	Neg	NC	NC	NC	NC
Monforte M [20]	19/f	Fe, Ph	Dy, Ga	NC	Neg	25	29	58	Neg
Pinto J [21]	8/m	Fe, Fa	Dr, He, Se	Slow bihemispheric activity, paroxystic activity on right frontocentral region	T2: increased signal basal ganglia, FLAIR: hyperintense signal of the caudate and lenticular nuclei. Edema in the cerebral cortex	76	118	3	Neg on 1st day, pos on 25th day
Rodrigo-Armenteros P [22]	14/m	Fe, Ph	Dy, Hy, A, decerebrated posturing	Anterior bilateral electrical status	Bilateral swelling basal ganglia	19	normal	normal	Pos
Grillo E [23]	13/m	Fe, Ph, pain limb	Ga, He, Se, aphasia	NC	Multifocal cortical signal abnormalities	2	48	93	Pos
Hongbo C [24]	21/f		He, Se	Abnormal but not specific waves	Multiple diffuse T2, FLAIR, and DWI in multiple regions				Pos
Van Lierde A [25]	9/f		A, Cr, He, photophobia, diplopia	Slight diffuse slowing	Cerebellar swelling, downward displacement of cerebellar tonsils, enlargement of lateral third ventricles	72	NC	NC	Neg
Sabat S [26]	24/f		He, Na, photophobia, Neck stiffness	NC	Cerebellar hemorrhage with intrinsic T1 hyperintensity	295	341	57	Pos
Yamashita S [27]	26/f	Fe, dyspnea, H, S	A, N	NC	NC	154	180	42	Neg
Ahn SW [28]	20/m	Fe, myalgia, H, S	He, D, Na, confusion, Se	NC	Signal lesion in the splenium of the corpus callosum	35	NC	NC	Pos
Dagdemir A [29]	8/f	V, Fe, L, H, S	He, Na, bilateral papilledema	NC	Symmetric hyperintensities on frontal white matter, capsula externa, and nucleus dentatus; effacement of cerebellar sulci; narrowing of fourth ventricle secondary to edema.	0	44	NC	Pos
Geurten C [30]	7/m	V, R, Bradycardia	Al, He, Neck stiffness	NC	Hyperintense swelling of the left cerebellar hemisphere with a mass effect on adjacent structures and, consequently, hydrocephalus	47	NC	NC	NC
Caruso JM [31]	9/m		A, He, Dy, lethargy	NC	Abnormalities with left superior temporal gyri enlargement and vascular enhancement consistent with gyral swelling	0	117	62	NC
Caruso JM [31]	11/f		He, Al, Dy, dystonic posturing	NC	Focal area of an abnormal, increased signal intensity in the subcortical white matter of the right cingulate gyrus isthmus with slight mass effect on the atria of the right lateral ventricle	23	122	33	Pos
Caruso JM [31]	17/m		Al, hyperreflexia	NC	Abnormal signal in the left frontal crowded radiated and a smaller focus in the right frontal subependymal white matter	44	246	32	Pos
Caruso JM [31]	5/m		Aphasia, Al, Se	NC	Symmetric swelling and increased signal intensity in the corpus striatum with mass effect on the frontal horn of the right lateral ventricle.	7	112	51	Pos
Caruso JM [31]	6/f		Al, V	NC	Increased signal intensity in dorsal pons and peridentate white matter of the cerebellum	350	102	32	Pos
Takeuchi S [32]	20/m	Fe, Diarrhea	He, confusion, neck stiffness	NC	T0 axial T2-weighted brain MRI: focal area of increased signal in the right temporal lobe; on day 2, axial T2-weighted brain MRI: progression of the lesion, with partial hemorrhagic conversion, acute brain swelling, and severe midline shift	14	160.7	99	Pos
Chadaide Z [33]	24/f	Fe	He, Se	Diffuse, bilateral slowing of the background activity with periodic lateralized epileptiform discharges	11th day: swelling of the gray matter in the inferio-mediotemporal-insular region, bilaterally	85	52	60	Pos
Biebl A [34]	12/m	Fa, Fe, ∗	He, Na, Se	NC	T0: no lesion. After 19 days (death of patient): focal lesions of increased signal intensity, edema in the cerebellum, brainstem, basal ganglia and hippocampus. Sagittal T1-weighted image: generalized cerebral edema with herniation of the brainstem and cerebellar tonsils into the foramen magnum.	43	93	90	Pos
Kim SY [35]	25/m	Fe, Ph	Numbness, paresthesia, GBS	NC	normal	4	102	55	NC
Kalita J [36]	6/m	Fe, conjunctival hemorrhage	Hallucination, Dr, decerebration, spasticity, hyperreflexia	Delta slowing	T2 hyperintense bilateral signal in the parieto-occipital region	155	64.6	43	Pos
Kalita J [36]	11/f	Fe, V	Se, extensor posturing, Cr	Theta slowing	Parieto-occipital lesions; bilateral on T2 and FLAIR images	10	37	57	

Abbreviations: A: ataxia; Al: altered behavior; Cr: cranial nerve palsy; D: dizziness; Dy: dysarthria; Dr: drowsiness; En: encephalitis; EBV, Epstein–Barr virus; Fe: fever; Fa: fatigue; FLAIR, fluid-attenuated inversion recovery; Ga: gait imbalance; GBS: Guillain–Barré syndrome; H, hepatomegaly; He: headache; Hy: hypokinesia; In: increased; L: lymphadenopathy; LV: left ventricle; N: nystagmus; Na: nausea; Neg: negative; NC: not checked; O: otalgia; Ph: pharyngitis; Pos: positive; R: rash; RV: right ventricle; S: splenomegaly, Se: seizures; V: vomiting; *: died.

## Data Availability

All relevant data are reported in the article. Additional details can be provided by the corresponding author upon reasonable request.

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
