# Peer review of "Post-Infectious Acute Cerebellar Ataxia Treatment, a Case Report and Review of Literature"

_children, 2023, doi:10.3390/children10040668_

Round 1

Reviewer 1 Report

Dear Prof. Dr. Paul R. Carney, Editor-in-Chief of Children Journal,

Dear Managing Editor of Children Journal,

Dear Authors of the Case Report Article,

January 28th 2023

RE: PEER-REVIEW REPORT – MINOR REVISIONS

       The article's authors (children-2210087) are discussing a case of acute cerebellar ataxia following an episode of infectious mononucleosis (IMN). The case report conveys essential information, yet anecdotal evidence, concerning the potential importance of intravenous immunoglobulin (IVIG) in managing the former condition. However, two main critical issues jeopardize the article's scientific value: the causality assessment between the assumed cause (acute IMN-induced cerebellitis) and the outcome (ataxia). At the same time, the authors also mentioned another potential causative agent (HPV vaccination) that might cause ataxia as part of an adverse event following immunization (AEFI), which is a well-known phenomenon to pharmacovigilance specialists. The other minor issue is the supplementary video material, and the authors should provide written informed consent dedicated to sharing that video and another permission for sharing the case details. Further, the English language of the manuscript requires moderate proofreading and revisions concerning some points detailed in the peer review report, and the authors should comply with each element carefully. I have also attached the original PDF submission file with highlights and comments to complement the peer review report, and the authors should study and correspond to each carefully.

·       The title is a good representative of the article and its contents

·       There are too many authors for a case report. Please refer to the journal's instructions for authors at https://www.mdpi.com/journal/children/instructions

·       The authors should refer to the journal's instruction of authors section concerning writing the abstract, which provides these details: "Abstract: The abstract should be a total of about 200 words maximum. The abstract should be a single paragraph and should follow the style of structured abstracts, but without headings: 1) Background: Place the question addressed in a broad context and highlight the purpose of the study; 2) Methods: Describe briefly the main methods or treatments applied. Include any relevant preregistration numbers, and species and strains of any animals used. 3) Results: Summarize the article's main findings; and 4) Conclusion: Indicate the main conclusions or interpretations. The abstract should be an objective representation of the article: it must not contain results which are not presented and substantiated in the main text and should not exaggerate the main conclusions."

·       The abstract's font size varies from one subsection to the other. The abstract section is good, although it requires some revision, proofreading, and editing per the journal's guidelines.

·       The authors should use the word "female" or "adolescent female" instead of "girl" for more professional medical writing.

·       The authors should not repeat keywords from the title, such as "acute cerebellar ataxia". The authors can revise the keywords based on the PubMed Medical Subject Headings (MeSH) and Emabase Emtree terms.

·       Concerning the keywords, the authors should consider revising "mononucleosis" to "infectious mononucleosis".

·       The introduction section is good. It conveyed background information, what is unknown about the subject of interest, and the aim of the study (case report). However, it is short, and the authors should provide additional citations and arguments from the literature and moderately increase the volume of the discussion section.

·       The case presentation section is excellent, unlike the conclusions section, which is inconsistent and does not abide by scholarly writing standards. The authors must revise the conclusions and introduction sections with high competency.

·       Line #76. How can the author exclude HPV vaccination as the causative factor leading to ataxia? The authors should provide a good argument based on one of the causality assessment criteria, such as the WHO-UMC criteria, available at https://www.who.int/publications/m/item/WHO-causality-assessment. The authors should also refer to these two relevant studies concerning adverse drug reactions (ADRs) and adverse events following immunization (AEFI):

·       Shukla AK, Jhaj R, Misra S, Ahmed SN, Nanda M, Chaudhary D. Agreement between WHO-UMC causality scale and the Naranjo algorithm for causality assessment of adverse drug reactions. J Family Med Prim Care. 2021; 10(9): 3303-3308. doi: 10.4103/jfmpc.jfmpc_831_21. PMID: 34760748; PMCID: PMC8565125.

·       Al-Imam A, Sami A, Lane S, Younus M. A concept for causality assessment and causal inference of adverse events cases. Journal of Biological Research. 2022; 95(2): 10772. doi: 10.4081/jbr.2022.10772

·       Line #97. How would the author interpret the coexistence of ataxia and hyperreflexia, indicative of upper motor neuron lesions (UMNL)? Is it due to generalized or localized encephalitis, perhaps affecting the motor? Can the premotor cortical area pathologies produce such UMNL manifestations?

·       Line #98. What is the significance or the interpretation of these EEG findings? The authors should explain to non-specialist audiences and readers who are not specialized in neurology or electro-physiology.

·       Line #103. The supplementary video is excellent and conveys interest to the article's readers. However, providing written informed consent concerning the video to the journal's editor is crucial.

·       Line #108. Please check the linguistic accuracy of the term "infectivologist" while proofreading.

·       Line #120. The unfortunate female patient remained hospitalized for almost 4 weeks. It would be interesting to the readers of your case to have a simple idea concerning the quality of life (QoL) of the patient and her family, the status of distress, and how the situation changed following the IVIG protocol administration.

·       The discussion section is the weakest within the full-text article. The authors focused mainly on findings within their reported case while not providing a relevant narrative based on the systematic review of the literature other than citing Table 1. Again, the authors should refer to guidelines for writing a good discussion section, in which they summarize the case report and the most significant findings while providing their interpretations and comparing those findings to other studies. The authors should also highlight potential limitations within their case study and recommendation for future research. It is not sufficient to cite the table, but they should also link the text narrative to table #1 while emphasizing the essential relevant cases and studies from the published literature.

·       Line #138. There are skipped citations. The authors jumped from citing reference #20 to reference #34. Where are the missed citations? The authors should check each in-text citation within the full-text manuscript very carefully.

·       The conclusions section is numeration is wrong.

·       The conclusions section is the weakest section of the article. It contains improper bibliographic citations while providing conclusions that cannot be extrapolated from a single case report. The authors should moderate the tone of the conclusions, proofread the section, and revise it according to the proper guidelines.

·       There should be no in-text citations within the conclusions section. The authors should revise with caution while referring to the journal's guidelines for authors.

·       This statement, "we might consider appropriate that, patients without no improvement of acute cerebellar ataxia should 6 be treated with intravenous immunoglobulin therapy.", should be moderated because the authors are reporting a single case (anecdotal evidence) per the hierarchy of the level of evidence. The former conclusions cannot be inferred with adequately designed randomized clinical trials (RCTs) and meta-analytic studies.

·       Concerning the authors' contributions, a case report should not have almost twenty authors. The journal's editor might decide against it. Please refer to the journal's guidelines concerning case reports. The excessive number of authors also made it extremely difficult to explore the existence of inappropriate self-citations within the references section.

·       The authors should provide a copy of the informed consent to the journal's editor, especially concerning the supplementary video material. Otherwise, the video should be retracted.

Thank you.

Best regards,

The peer-reviewer.

Author Response

Thank you

Dr Emanuela Del Giudice

Reviewer 2 Report

I would like to start by congratulating the authors on their article. The authors report on a young adolescent with para/post infectious ataxia, and discuss not only the investigation performed, but also treatments and perform a literature review. The video is much interesting and adds value to the article

The case is interesting, as well as the literature review. I would just have a few comments:

-        Introduction: On introduction the authors state that an acute cerebellar ataxia usually develops after a viral illness. It is one of the etiologies of acute ataxias, but we have many others (vascular, paraneoplastic, intoxication). It should be rephrased

-        Case report: On lines 60-70 the authors state that “As the patient was examined during the pandemic and neurological symptoms are also present in SARS-infected children, antigenic and molecular swabs for SARS-CoV-2 were performed resulting negative”. At this time of the case description there were no neurological symptoms. Perhaps there was another reason for performing SARS Cov 2 test

-        Case report: on page 3 the authors describe the patient was started on acyclovir. For how long? It would be better to have a comment on discussion on the reason for starting acyclovir even with a negative PCR for HSV

-        Case report: What is the duration of the follow up now?

-        Discussion: I understand the authors are paediatricians and writing to a paediatric journal, but acute cerebellar ataxia does not occur only in children. The first sentence should be rephrased

-        Discussion: The discussion on whether ataxia was related to vaccination or to EBV should be more detailed. Actually, both may trigger ataxia, treatment strategies are the same, and the reasons for considering this case as post EBV should be more clear.

-        Discussion: The tables are much detailed, but a summary of the previously described cases, clinical findings, exams should also be performed on the text.

Author Response

Thank you 

Dr. Emanuela Del Giudice

Round 2

Reviewer 2 Report

I would like to start by congratulating the authors on their case report. I believe all the reviewers concerns were correctly addressed.